# Auditory cortical alpha/beta desynchronization prioritizes the representation of memory items during a retention period

Nathan Weisz*, Nadine Gabriele Kraft, Gianpaolo Demarchi*

Centre for Cognitive Neuroscience and Department of Psychology, Paris-Lodron Universität Salzburg, Salzburg, Austria

**Abstract** To-be-memorized information in working-memory could be protected against distracting influences by processes of functional inhibition or prioritization. Modulations of oscillations in the alpha to beta range in task-relevant sensory regions have been suggested to play an important role for both mechanisms. We adapted a Sternberg task variant to the auditory modality, with a strong or a weak distracting sound presented at a predictable time during the retention period. Using a time-generalized decoding approach, relatively decreased strength of memorized information was found prior to strong distractors, paralleled by decreased pre-distractor alpha/beta power in the left superior temporal gyrus (lSTG). Over the entire group, reduced beta power in lSTG was associated with relatively increased strength of memorized information. The extent of alpha power modulations within participants was negatively correlated with strength of memorized information. Overall, our results are compatible with a prioritization account, but point to nuanced differences between alpha and beta oscillations.

**\*For correspondence:**
nathan.weisz@sbg.ac.at (NW);
gianpaolo.demarchi@sbg.ac.at
(GD)

**Competing interests:** The authors declare that no competing interests exist.

## Introduction

Adaptive sensory processing entails the prioritization of task-relevant features with respect to competing information. Top-down modulation of activity in neural ensembles encoding task-relevant or distracting information is crucial in achieving this goal. In particular, regionally specific power changes around the alpha frequency range have been linked to such a putative top-down-mediated gain modulation, with enhanced power reflecting relatively inhibited states (*Jensen and Mazaheri, 2010*; *Klimesch et al., 2007*). For the visual modality especially, a vast amount of empirical evidence supports this notion. For example, increased alpha power in parieto-occipital cortical regions contralateral to the unattended hemifield is a very robust finding (e.g. *Busch and VanRullen, 2010*; *Thut et al., 2006*). The general inhibitory gating function of localized alpha increases has also been reported with respect to more specific visual features, leading to remarkable spatially circumscribed alpha modulations (*Jokisch and Jensen, 2007*; *Zumer et al., 2014*) even at a retinotopic level (*Popov et al., 2019*). Also for the domain of working memory, alpha *increases* have been reported during the retention period in the visual (e.g. *Jensen et al., 2002*; *Klimesch et al., 1999*), somato-sensory (e.g. *Haegens et al., 2009*) and auditory modalities (e.g. *Obleser et al., 2012*), putatively protecting the to-be-remembered information against interference. This load-dependent top-down amplification of alpha and its concomitant inhibition account are widely accepted, but circumscribed *decreases* in alpha to beta power (often labeled as desynchronization) have also been deemed functionally important in the context of working-memory tasks. In a prioritization account, they reflect an enhanced activation of performance-relevant neural ensembles (e.g. *Noh et al., 2014*; *Sauseng et al., 2009*). A recent framework by *Hanslmayr et al., 2016* explicitly links the extent of

alpha/beta desynchronization to the representational strength of the information content in episodic memory (for supportive evidence see *Griffiths et al., 2019*). This is in line with a framework by *van Ede, 2018* stressing the importance of regionally specific alpha and beta decreases when item-specific information needs to be prioritized in the retention period of working-memory tasks.

Distracting sounds are ever-present in natural listening environments and necessitate flexible exertion of inhibition or prioritization processes. Besides stimulus-feature information, which can influence the precise location of alpha modulations in the visual system (*Popov et al., 2019*), temporal cues can also be exploited (*Rohenkohl and Nobre, 2011*; *van Ede and Chekroud, 2018*): that is, when distracting sound input can be temporally predicted, inhibition or prioritization processes should be regulated in an anticipatory manner in relevant auditory regions. As in other sensory modalities (*Frey et al., 2015*; *Weisz and Obleser, 2014*), an increasing amount of evidence points to a functional role of alpha oscillations in listening tasks. Increased alpha oscillations have been observed in putatively visual brain regions when focusing attention on auditory input in cue-target periods (*Frey et al., 2014*; *Fu et al., 2001*; *Snyder and Foxe, 2010*). A similar posterior pattern is also observed in challenging listening situations, for example, with increased cognitive load or when faced with background noise (for reviews see *Johnsrude and Rodd, 2016*; *Rönnberg et al., 2011*). However, increases in alpha oscillations as a mechanism for selective inhibition (*Strauß et al., 2014*) have rarely been shown for auditory cortex, in which feature-specific processing of target and distractor sounds takes place. With regards to alpha desynchronization in auditory cortex, different lines of evidence showing an association between (also illusory) sound perception and low auditory cortical alpha power (e.g. *Lange et al., 2014*; *Weisz et al., 2007*; *Weisz and Obleser, 2014*; for invasive recordings illustrating sound-sensitive alpha desynchronization in anterolateral Heschl's Gyrus, see *Billig et al., 2019*), suggest a link to representational content as described above.

The goal of the present study was to test whether power modulations in the alpha/beta range (*Hanslmayr et al., 2016*; *van Ede, 2018*) in task-relevant auditory cortical areas prior to a temporally predictable distractor, which was presented in the same (i.e. auditory) modality as the target, would better fit with an inhibition or prioritization account. On a general level, power increases would be predicted by an inhibition account, whereas decreases would be expected according to a prioritization account. Furthermore, both alternative accounts make opposing predictions regarding the relationship between pre-distractor alpha modulations and the strength of memorized information in the retention period (see *Figure 1*).

We adapted a Sternberg task variant introduced by *Bonnefond and Jensen, 2012* to the auditory modality. These researchers illustrated pronounced alpha and beta increases, as well as phase effects, in parieto-occipital regions prior to the presentation of a more potent but temporally predictable visual distractor in the retention period. Using magnetoencephalography (MEG) and decoding, we first identified regions that were informative as to whether a speech item was part of a memory set or not, and focused subsequent spectral analysis on the left superior temporal gyrus (lSTG). This region, which is crucially involved in phonological short-term memory (*Jacquemot and Scott, 2006*), expressed marked alpha/beta desynchronization prior to a strong distractor. Importantly, by time-generalizing the aforementioned classifier (*King and Dehaene, 2014*), we implemented a proxy for the strength of memorized information that could be compared between trials with high or low power. Specifically, we show that lower pre-distractor beta power in lSTG goes along with relatively enhanced memory representation in the same period. For alpha power, however, a negative correlation was observed between the strength of memorized information and the extent to which power was modulated at an individual level. Overall, our study draws a nuanced picture that points to differential alpha and beta processes in the auditory cortex that altogether support the prioritization of relevant information in working memory (*van Ede, 2018*).

## Results

Thirty-three healthy participants performed a modified Sternberg task (*Bonnefond and Jensen, 2012*) adapted to the auditory modality. In each trial they listened to a sequence of four consonants spoken by a female voice (see *Figure 1A*). These items had to be memorized across a 2-s retention period, after which a probe item was presented. Participants were requested to report whether the probe item was part of the memory set or not. Critically, precisely 2 s after onset of the last memory item, a distractor was presented that was either strong (a consonant spoken by a male voice) or

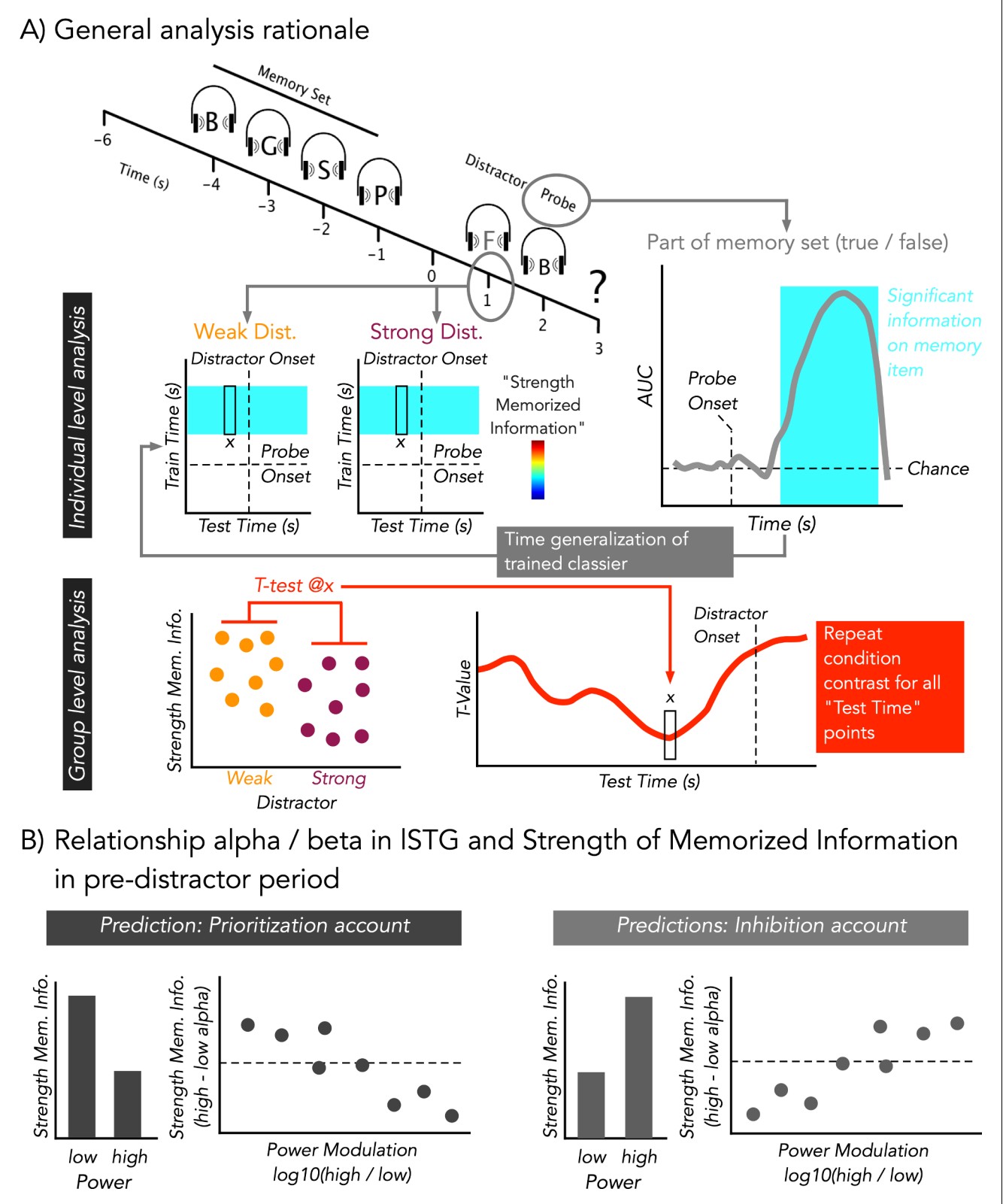

**Figure 1.** Modified auditory Sternberg paradigm and cartoon depiction of analysis rationale. (**A**) A sequence of four consonants spoken by a female voice was presented. After the retention period, either a strong (consonant spoken by a male voice) or a weak (scrambled consonant) distractor was presented (at 1 s). Distractor type was kept constant during a block. Subsequently, participants indicated by a button press whether the probe was part of the memory set ('part') or not ('no part'). At an individual level, temporal decoding was performed on whether the probe was part of memory set or

*Figure 1 continued on next page*

*Figure 1 continued*

not. When the probe was part of the memory set, it should have been seen to share distinct neural patterns with those elicited by the items of the memory set, while this should not have been the case when the stimulus was not part of the memory set. By time-generalizing the classifiers trained on the probe to the period of the retention interval, we obtained a quantitative proxy for the strength of memorized information at the time of distractor presentation. The results were then statistically contrasted between weak and strong distractors across the group. (B) Alpha/Beta power in lSTG was calculated at a single trial level in a pre-distractor period and was used to bin high and low power trials. For a 0.5-s pre-distractor period, analysis analogous to (A) was performed to quantify the relationship between regionally specific alpha/beta power and strength of memorized information. A prioritization account would predict that lower 'desynchronized' states go along with relatively increased strength of memorized information. This pattern should be captured when contrasting the bins across the entire group and when taking into account the extent of modulation within single participants. An inhibition account would predict an opposing pattern.

weak (an acoustically scrambled version of a consonant). The different distractor types were presented blockwise in a manner counterbalanced across participants.

## Adverse behavioral impact of strong distractors

We reasoned that the processing of a strong distractor would be more difficult to suppress and should affect behavioral performance. Comparing average accuracy between the strong and the weak distractor conditions showed a small (85% vs 83%) but statistically significant ($t_{32}$ = −2.11, $p_{one-sided}$ = 0.02; Cohen's d = 0.37, 'small effect') deterioration of performance for strong distractors. Reaction times were on average 9 ms slower for the strong distractor condition (579 ms vs 570 ms); however, this difference was not significant ($t_{29}$ = 1.07, $p_{one-sided}$ = 0.14; Cohen's d = −0.20, 'very small effect'). It should be noted that response speed was not emphasized to avoid interference from button presses on processing of the probe item. This may have reduced potential reaction time differences. Overall, the behavioral analysis supports the notion that the strong distractor condition was slightly more challenging, laying a solid foundation for the subsequent MEG analysis.

## Decoding probe-related information

Our main goal was to investigate the extent to which different distractor levels influence the strength of memorized information during the retention period, in particular prior to the predictable onset of the distractor. Also, we wanted to relate these effects to potential alpha power modulations in the pre-distractor period. To this end, we first applied temporal decoding, using linear discriminant analysis (LDA, 5-fold cross validation, repeated five times, AUC as a metric; see *Figure 1A* and 'Materials and methods' for details), on the post-probe MEG sensor-level activity, to classify whether a probe was part of the memory set or not. The results are depicted in *Figure 2A* and show robust and sustained above-chance classification performance rapidly commencing ~334 ms after probe onset ($p_{cluster}$ = 4e-4) and lasting until the onset of the response prompt at 700 ms post-probe (Cohen's d in this period was generally >0.8, that is 'strong and very strong effects'). The time course of this effect is very much in line with those seen in evoked response studies on old vs new effects in short-term memory (*Danker et al., 2008*; *Kayser et al., 2003*), indicating that early sensory activation is not informative on whether a probe was a memorized item or not. In subsequent analyses, a 0.4–0.7-s post-probe decoding period was used as a training set, and the derived classifiers were applied in a time-generalized manner to the preceding retention period (yielding a quantitative proxy for the strength of memorized information; see below and also *Figure 1*).

Although the results so far show that whether a probe was part of a memory set or not can be differentiated based on the MEG data, the resulting temporal decoding pattern uses all sensors and is therefore spatially agnostic. In order to obtain insights into which brain regions may be contributing to the effect and aiding the identification of task-relevant region(s) in a data-driven manner (*Jacquemot and Scott, 2006*), we adapted an approach to derive Informative Activity in source space (*Marti and Dehaene, 2017*). In brief, this approach projects the sensor level classifier weights to source space using beamformer filters. To make the effects more interpretable, we implemented a within-subject permutation analysis and z-scored the classifier-weights in a first-level analysis. These data were subsequently tested at group-level against zero within a nonparametric cluster permutation test using a *t*-test, yielding a positive cluster (p=4e⁻⁴) collapsed over the relevant post-probe time period. As shown in *Figure 2A*, Informative Activity can be detected in widespread cortical regions encompassing temporal, parietal and frontal areas. Although the pattern was bilateral,

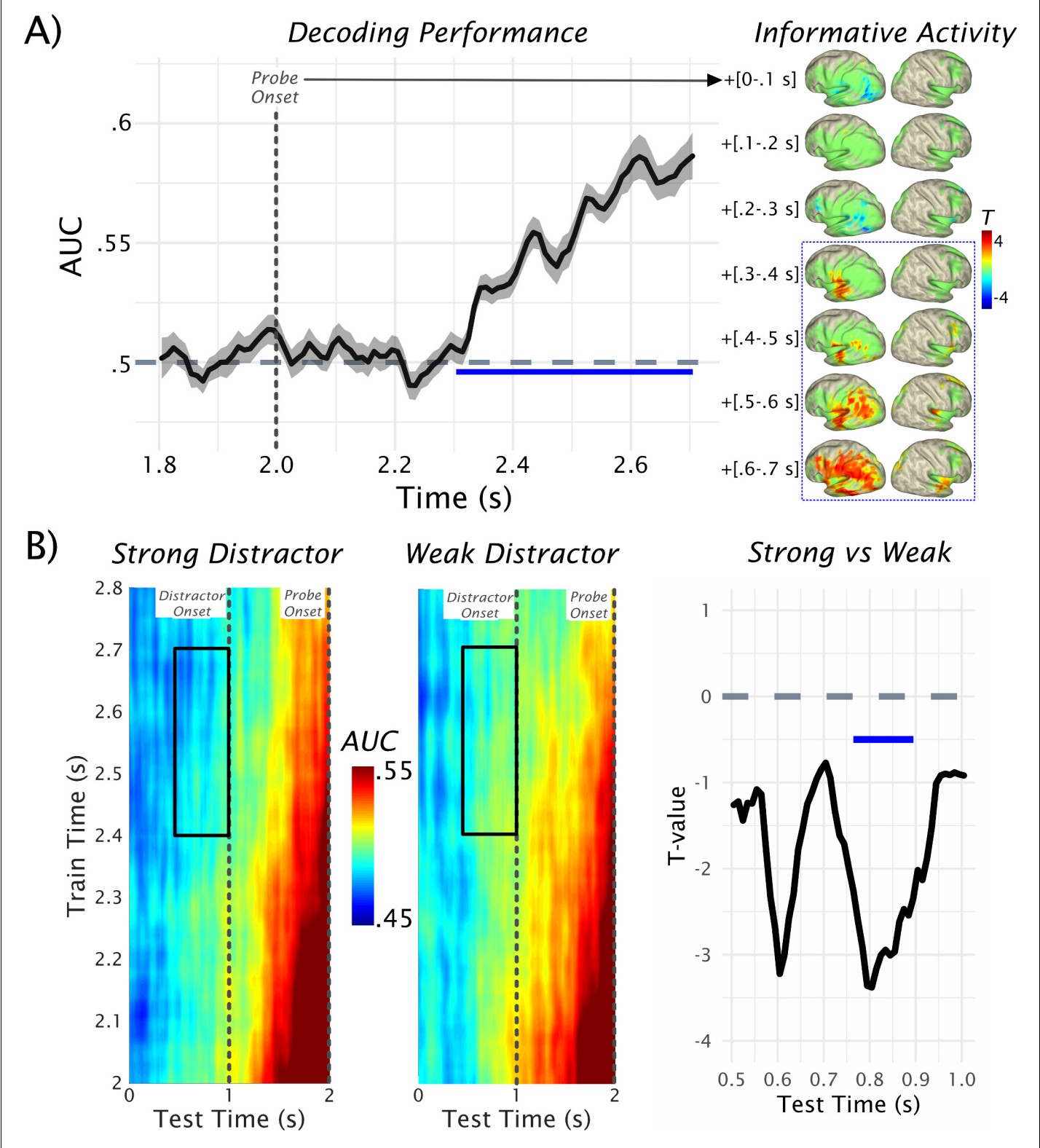

**Figure 2.** Decoding of probe-related information. (**A**) Results of the temporal decoding of MEG sensor-level activity after the probe presentation to classify whether it was part of the memory set or not (see also *Figure 1A* for rationale). Above-chance detection performance (AUC = area under the receiver operating characteristic [ROC] curve) was found commencing ~300 ms after the probe onset (at 2.0 s) and lasting at least until the response was prompted. Informative activity for this decoding as a function of time is shown on the right (green areas outlining the expanse of the cluster

*Figure 2 continued on next page*

*Figure 2 continued*

that results following a nonparametric permutation test for the 0.3–0.7-s post-probe onset interval). Within the relevant time interval (in blue-dotted box) informative activity emerges early in left STG and progressively spreads to further temporal, parietal and frontal areas. Data used for plotting the results of the temporal decoding at 10.17605/OSF.IO/753MK. (B) The time generalization result is shown separately for the strong and weak distractor conditions (left and middle panels). Trivially, the strongest classification results are obtained approximately at the onset of the probe (at 2 s). Relatively decreased decoding performance (AUC <0.5) was obtained prior to the onset of the strong distractor. Statistical comparison of strong vs weak distractor conditions revealed two peak effects at ~400 ms and ~200 ms preceding the distractor onset, although only the difference closer to distractor onset was significant ($p_{cluster}$ = 0.0156) at the cluster level (right panel). Data used for plotting the time generalized result at 10.17605/OSF.IO/4CV83.

there was a clear left-hemispheric dominance, with the most pronounced effect localizable to lSTG. Given the particular involvement of this latter area in processing speech sounds (e.g. *Mesgarani et al., 2014*; see also *Billig et al., 2019*) and its central role in phonological short-term memory (*Jacquemot and Scott, 2006*), it was used as a task-relevant region of interest for the spectral analysis (see below).

After illustrating the above-chance decoding performance for the probe, we tested the extent to which informative patterns are present in the retention period, and how they are modulated by the distractor especially prior to its anticipated onset. The classifier described above was employed for this purpose using time generalization (*King and Dehaene, 2014*). The rationale for this approach (see also *Figure 1*) was that if a probe was part of the memory set, it should share neural patterns with those elicited by the actual memory set. This should not be the case for probes that were not part of the memory set, so that time-generalizing these post-probe patterns to the retention period should give a quantitative proxy for the strength of memorized information. The full time generalization result is shown separately for the strong and weak distractor conditions in *Figure 2B*. Trivially, the strongest classification results are obtained approximately at the onset of the target (at 2 s), which corresponds to the area close to the diagonal of the time generalization matrix. We were mainly interested in the decoding performance in the period preceding the anticipated distractor at 1 s, that is, in the off-diagonal pattern. It is evident that for the late 0.4–0.7-s post-probe training time period (probe onset at 2 s shown on the y-axis), decoding accuracy gradually ramps up over the retention interval.

Statistical analysis was done for a 500-ms window preceding the onset of the distractor (*Figure 2B*, right panel), focusing on the data-driven 0.4–0.7-s post-probe training time window (see above). This analysis yielded two peak effects at ~400 ms and ~200 ms (Cohen's d in these periods ~0.6 and ~0.55, respectively, that is, medium effect sizes) preceding the distractor onset, during which critical *t*-values (±2.0369) were exceeded. However, only the latter difference was significant following a nonparametric permutation test ($p_{cluster}$ = 0.0156). These results suggest that memorized information is differentially activated prior to distractor onset, with relatively reduced activation prior to the strong distractor. In fact, when post-hoc testing the decoding accuracy over the entire test- and training-time window, average decoding accuracy prior to the strong distractor was significantly below chance ($t_{32}$ = −3.98, p=3.63e-04), whereas it did not differ for weak distractors ($t_{32}$ = −1.73, p=0.09). Below-chance decoding may appear surprising but is not uncommon in time- and condition-generalization decoding approaches (*King and Dehaene, 2014*).

Although the precise neural processes leading to these patterns are challenging to pinpoint, in functional terms, this result translates into a *systematic* relative absence of memory-item specific patterns akin to those elicited during relevant periods following probe onset. As this activation gradually ramps up toward the anticipated onset of the probe in both conditions, albeit somewhat delayed in the strong distractor condition, it is clear that some representation with regards to the memorized item would also need to be present during the periods of below chance decoding. Recently, *Stokes, 2015* and *Wolff et al., 2017* described network-level 'activity silent' processes encoding working memory content, and similar processes could be present in the early part of the retention period in the present task. Overall, the described decoding effect is in line with the behavioral results described above, implying an overall detrimental impact associated with anticipation of a strong distractor.

## Pre-distractor modulations of induced oscillatory activity

In a next step, we focused on modulations of oscillatory activity in the lSTG, a region dominating our source level analysis of informative activity with regards to the probe decoding (i. e. whether memorized or not) and strongly suggested to be task-relevant for phonological short-term memory by previous research (see, for example, *Jacquemot and Scott, 2006*). We focused our (statistical) analysis on the period immediately preceding the predictable occurrence of the distractor. According to an inhibition account, alpha/beta enhancements would be expected to precede the strong distractor in particular, putatively reflecting an anticipatory suppression of its processing. A prioritization account, on the other hand, would predict a power reduction in the same frequency range, putatively reflecting an anticipatory activation of the memorized information.

The time-frequency representations in *Figure 3A*, which display the induced power in the 5–25 Hz range, show strong ongoing alpha/beta activity with a peak ~10 Hz in the lSTG. Analogous to the study by *Bonnefond and Jensen, 2012*, a 500-ms period preceding the occurrence of the distractor is marked, suggesting an alpha power decrease in the strong as compared with the weak distractor condition. This impression is supported by a nonparametric permutation test ($p_{cluster}$ = 0.0104), which yields a significant difference in this period with peak differences of ~12–13 Hz and ~21–22 Hz (*Figure 3A*, right panel), comparable to those recorded in the study in the visual domain (*Bonnefond and Jensen, 2012*). Given the perfect temporal predictability of the distractor occurrence, stronger prestimulus phase alignment of alpha oscillations could be expected (as reported in the visual modality by *Bonnefond and Jensen, 2012*; but see *van Diepen et al., 2015*). This process putatively exploits the fact that excitability varies over an alpha cycle, to align its inhibitory phase optimally to suppress processing of the irrelevant sound maximally. However, even though clear post-distractor evoked alpha enhancements could be observed (see *Figure 3B*), no prominent evoked alpha could be observed preceding the distractor (an analysis using ITC leads to an identical conclusion; data not shown). For the sake of completeness, we ran an analogous statistical test as for the induced power, showing no difference at the cluster corrected level (*Figure 3B*, right panel). Since no pronounced evoked alpha activity was identified in the pre-distractor period, we refrained from further analysis (such as phase opposition effects). This result extends a previous report (*van Diepen et al., 2015*) in finding no evidence that auditory cortical alpha phase is adjusted in a top-down manner.

Overall, the spectral analysis suggests a pronounced decrease of alpha to beta power prior to the expected occurrence of the strong distractor. Although this pattern appears to support a prioritization account, it does not fit well with the decoding result at a first sight. One interpretation, along the lines of the inhibition account, could be that the expectation of a more salient auditory distractor may involuntarily draw more selective attention towards it (reflected in an anticipatory alpha decrease), making it more difficult to suppress. On the basis of the results presented so far, however, these alternatives cannot be differentiated. In the next part, by exploiting the possibility of time-generalizing the classifiers trained above (for rationale see also *Figure 1A*), we will attempt to address this important issue by linking pre-distractor alpha power modulations to the strength of memorized information in the retention period.

## Strength of memorized information and pre-distractor alpha power

To address the functional relevance of pre-distractor alpha and beta power modulations in the lSTG in greater detail, trials were sorted according to alpha (13 ± 3 Hz) or beta (21 ± 3 Hz) power in this region in a 400–1000-ms time period following the onset of the retention period (i.e. a 600-ms pre-distractor window centered on peak latency effect at 700 ms). Subsequently, these trials were median split into high and low alpha or beta power bins. Analogous to the analysis described above (see also *Figure 2*), we trained a classifier on all trials to discriminate whether a probe was part of a memory set or not, and applied the classifier to a 0.5-s time-window prior to distractor presentation separately for the high- and low-power trials (analogous to the analysis shown in *Figures 2* and *3*) . On the basis of the previous analysis, we again focused on a 400–700-ms training time period and calculated the average strength of memorized information for each bin as well as the difference between bins for each individual participant. A functional relationship between pre-distractor alpha/beta power and strength of memorized information should be reflected in two ways (with different directions predicted according to a prioritization or inhibition account; see *Figure 1B*). First, strength

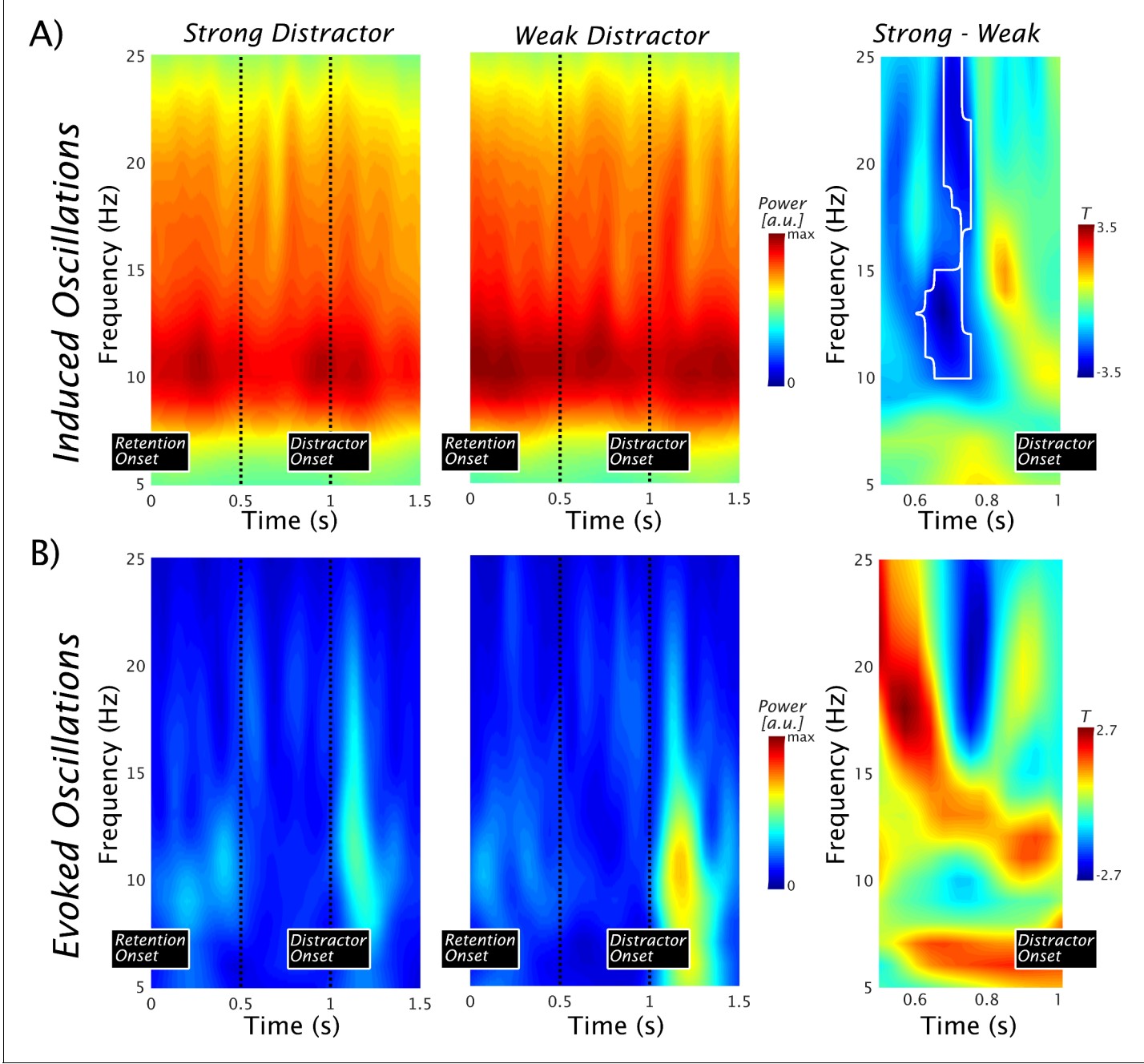

**Figure 3.** Pre-distractor alpha power modulations in the left superior temporal gyrus. (**A**) Time-frequency representations of the induced power show strong ongoing alpha/beta activity with a peak at ~10 Hz. No baseline normalization was applied. The vertical dots indicate a 500-ms period preceding the occurrence of the distractor. An alpha/beta power decrease in the strong vs the weak distractor condition can be seen (left and middle panel). The notion is supported by the outcome of a nonparametric permutation test leading to a significant difference at cluster level (marked by a black contour; $p_{cluster}$ = 0.0104) over an alpha to beta range with peak differences at ~12–13 and 21–22 Hz (right panel). For both ranges, the peak effects were observed at 0.7 s, that is, ~300 ms prior to the anticipated onset of the distractor. Data used for plotting induced power at 10.17605/OSF.IO/4WUYD. (**B**) Time-frequency representations of the evoked power. Post-distractor alpha enhancements are seen, but no prominent alpha preceded the distractor (left and middle panel). The nonparametric statistical test at cluster level showed no difference (right panel). Data used for plotting evoked power at 10.17605/OSF.IO/TYZC8.

of memorized information should differ *overall* between low- and high-power bins. Second, stronger (relative) differences between the bins should be reflected in stronger concomitant differences in the strength of memorized information.

To test for overall differences in the strength of memorized information between low- and high-power bins we first ran a repeated measures ANOVA using frequency band (alpha and beta) and bin (high and low) as factors. This analysis showed that the strength of memorized information did not differ overall between the frequency bands ($F_{1,32} = 1.19$, p=0.28). A trend was observed for bins ($F_{1,32} = 3.29$, p=0.08) with low-power bins showing relatively increased strength of memorized information (see *Figure 4A*). However, the interaction effect ($F_{1,32} = 4.06$, p=0.05) indicated that this difference was not uniform for the alpha and beta bands. Comparing strength of memorized information within each frequency band showed that a difference was only significant for beta ($t_{32} = 2.27$, p=0.03) and not for alpha ($t_{32} = 0.06$, p=0.95). Thus, in line with a prioritization account (see *Figure 1B*), overall pre-distractor power in the beta frequency range in lSTG goes along with relatively increased strength of memorized information.

The previous approach treats low- and high-power bins equally among participants. If there is a functional relationship between the oscillatory processes in lSTG and strength of memorized information, however, more extreme power differences between the bins should accompany more

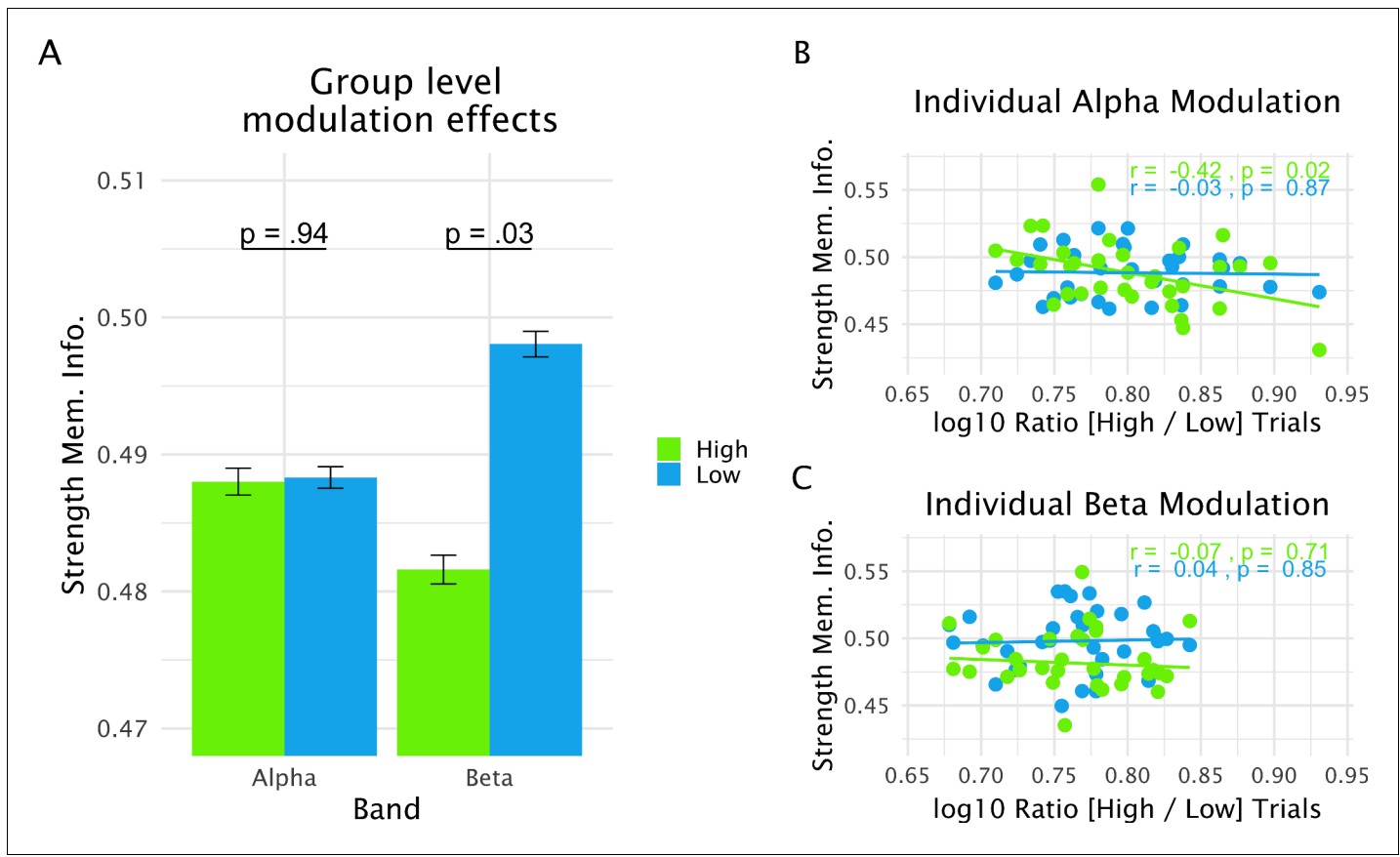

**Figure 4.** Relationship between alpha and beta power modulation and strength of memorized information (operationalized via the time-generalized decoding approach; see *Figure 1*) in the 0.5-s pre-distractor period. (**A**) Average strength of memorized information in the relevant period split between strong and weak power trials. At the group-level, significant modulation is seen only for the beta band, with relatively increased strength of memorized information for weak power trials. (**B**) Interindividual variation in the extent to which alpha power was modulated between high- and low-power trials within a participant (a higher value on the x-axis reflects a more extreme power difference between high- and low -power trials) was negatively correlated with the modulation of strength of memorized information (see main text). This effect was in large part driven by the strong power trials. (**C**) The same correlation analysis showed no effect in the beta band. Overall, power in the alpha range was more strongly modulated as compared to that in the beta range. Data used for plotting the relation between alpha power modulations and probe-related information is at 10. 17605/OSF.IO/QG3KB.

extreme differences in terms of strength of memorized information (see *Figure 1B*). In this respect, an interesting first observation is that modulations between low- and high-power trials (computed in each participant as the $\log_{10}$ ratio between high and low power trials in the relevant band) were significantly stronger for alpha oscillations as compared to beta oscillations ($t_{32} = 3.11$, p=0.004; compare also *Figure 4B and C*). Correlating this power modulation measure with the differences in strength of memorized information between low- and high-power trials yielded a significant effect only for alpha oscillations (alpha — $r = -0.36$, p=0.04; beta — $r = 0.06$, p=0.72), a pattern that better fits a prioritization account (see *Figure 1B*). A follow-up correlation analysis on the separate bins shows that for alpha this effect is driven by the high-power trials (*Figure 4B*). Altogether, relatively desynchronized states in the alpha and beta bands appear to go along with relatively increased strength of memorized information, which is in accordance with a prioritization process. Interestingly, our analysis of these trials points to a more nuanced and differential picture for alpha and beta oscillations, which have been frequently treated in a homogenous manner. Beta desynchronization appears to prioritize memorized information in general, whereas the alpha processes seem to be dependent on individually varying modulations.

## Memory-related information is mainly carried by low-frequency activity

So far we have established that overall alpha and beta power in lSTG is relatively reduced prior to the onset of an anticipated strong distractor, and that this process could be involved in prioritizing auditory information in working memory. A valid concern could be that if it were mainly alpha/beta power reductions in lSTG that were carrying the decodable information (trained post-probe onset and time-generalized to retention period) then this would make the analysis somewhat circular. Although this reasoning conflicts with the condition comparison effects showing that lower alpha/beta powers prior to strong distractors are associated with *on average* lower decoding accuracy for memorized information, we explicitly followed up this issue. For this purpose, the basic decoding shown in *Figure 1A*, that is whether a probe was part of a memory set or not, was repeated following filtering of the data in different frequency bands (broadband — 1–30 Hz; theta — 1–7 Hz; alpha — 9–17 Hz; beta — 18–24 Hz) and by either applying the Hilbert transform or not. Importantly for the purpose of addressing the potential circularity issue, decoding was poor for the alpha and beta bands in general (see *Figure 5A*), and did not differ from chance for the relevant time period upon which our time-generalized effects are based (see *Figure 5B*).

For the broadband analysis, decoding was also significantly above chance when using only the analytic amplitude part following the Hilbert transform ($t_{32} = 4.19$, p=0.0002), although this is significantly lower than when using the original broadband signal ($t_{32} = -6.22$, p=5.68e$^{-07}$). A similar pattern can be observed for the theta band, with significant above-chance decoding for the real ($t_{32} = 9.62$, p=5.78e-11) and analytic amplitude ($t_{32} = 6.26$, p=5.11e-07) versions of the signal (for real vs analytic amplitude, $t_{32} = -4.80$, p=3.25e$^{-05}$). Decoding when using only the theta band was overall superior as compared to that using the broadband signal (real signal — $t_{32} = 10.71$, p=4.10e$^{-12}$; analytic amplitude — $t_{32} = 7.11$, p=4.53e$^{-08}$). To summarize: alpha/beta band modulations, which are different on average prior to strong and weak distractors, do not appear to carry actual (decodable) information about the memorized item (see also *Griffiths et al., 2019*). The most important signal component contributing to our analysis (*Figure 1A*) is in the theta frequency range. Interestingly, the temporal fine structure, as compared to simple amplitude modulations, seems to contain relevant representational information.

## Discussion

In the current study, we investigated the neural dynamics prior to an anticipated distractor in the auditory modality. We were particularly interested in potential modulations of alpha power in auditory cortical regions, which have shown patterns similar to those described in various cognitive tasks in the visual system (*Frey et al., 2015*). Also alpha-power modulations, although mainly in non-auditory regions, have been previously linked to listening effort or attentional control (*McGarrigle et al., 2014*; *Pichora-Fuller et al., 2016*; *Wöstmann et al., 2017*). In order to understand the functional relevance of potential alpha-power modulations, it was important to link them to the informational content carried by the neural patterns in the same time period. For this purpose, we adapted a modified Sternberg paradigm first proposed by *Bonnefond and Jensen, 2012* to the auditory system:

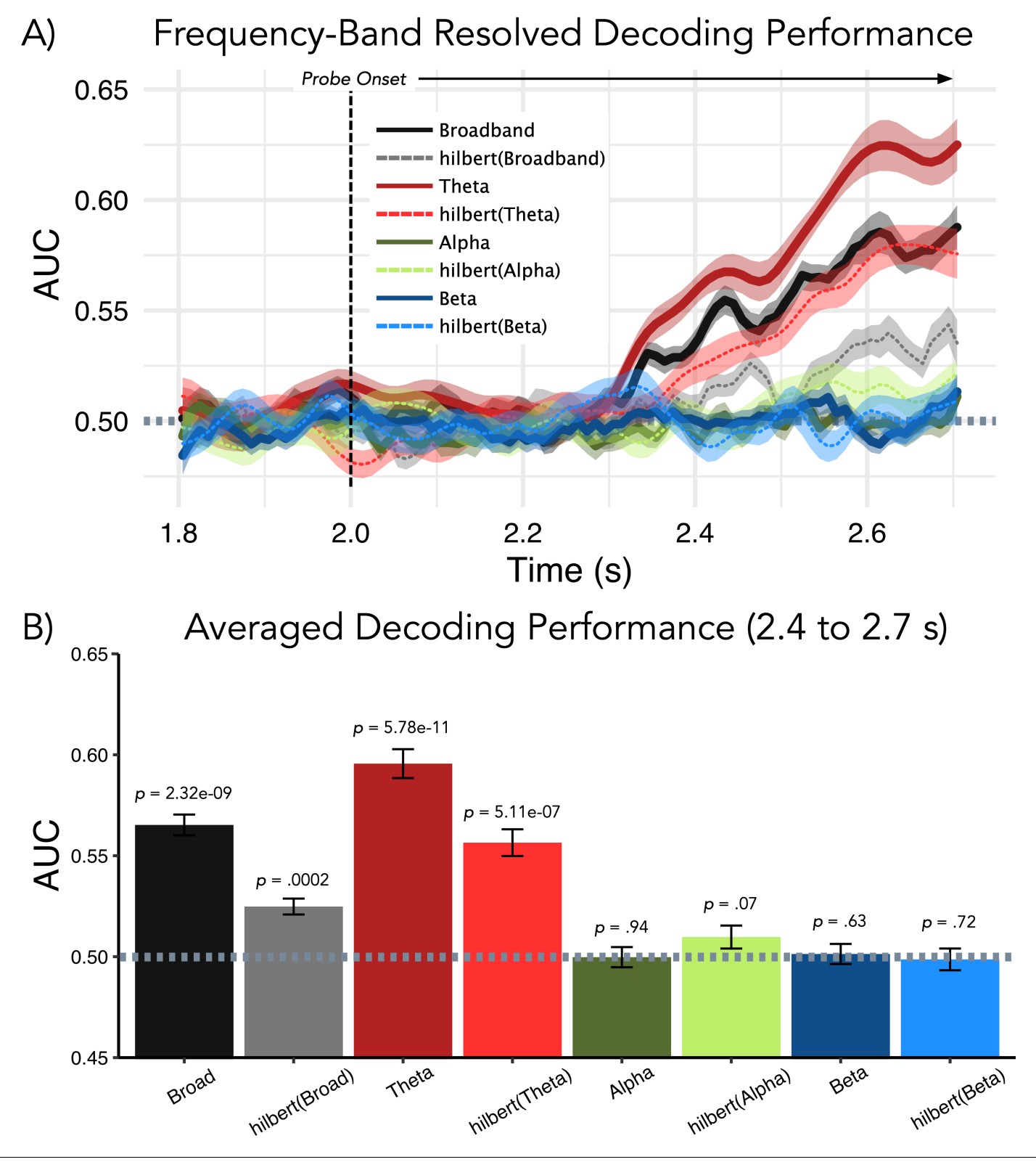

**Figure 5.** Follow-up analysis to elucidate which frequency band drives decoding performance post-probe presentation (used as trained classifiers time-generalized to retention period). (A) Results of temporal decoding on MEG sensor-level activity analogous to *Figure 2A*. (B) Average decoding performance for the relevant 0.4–0.7-s post-probe time period (used for results displayed in *Figure 2B* and *Figure 4*) shows that neither alpha nor beta band activity contains information with regards to the memorized item. As shown previously, above-chance decoding performance is seen for

*Figure 5 continued on next page*

*Figure 5 continued*
broadband activity and this effect appears to be driven strongly by activity in the theta range. Although amplitude information was sufficient for decoding above chance for broadband and theta activity, decoding performance was improved when the temporal fine-structure was maintained.

that is, we introduced (in a block wise manner) putatively weak and strong auditory distractor items in the retention period with predictable timing. Although perfect temporal predictability of a distractor is rare in natural environments (e.g. ticking of clock, dripping faucet), this was maintained in the present study to assure maximum comparability. In particular, strict temporal predictability should boost some potential effects, especially those pertaining to increasing pre-distractor phase consistency.

The behavioral effects are weaker than in the original visual experiment by *Bonnefond and Jensen, 2012*. Overall, the behavioral finding of lower accuracy for strong distractors suggests an adverse effect of strong acoustic distractors on the representation of memorized information. This notion was supported on a neural level using a time-generalization approach (*King and Dehaene, 2014*). This encompassed first training of classifiers to decode whether a probe item was part of the memory set or not, with these classifiers being used subsequently as a quantitative proxy for the strength of memorized information. These classifiers were then time-generalized to the period around the distractor presentation in the retention interval. Interestingly, classifier accuracy, especially prior to the strong distractor, was significantly below chance level. Such patterns are not uncommon in M/EEG studies using time- and condition-generalized decoding (see *King and Dehaene, 2014*) and are also seen in fMRI studies (e.g. *van Loon et al., 2018*). Descriptively, in electrophysiology, below-chance decoding can arise when the neural patterns underlying representations are opposing and/or temporally shifted. Thus below-chance decoding cannot be interpreted as the absence of condition- or feature-relevant information. However, a functional interpretation is challenging (*King and Dehaene, 2014*). On the basis that our approach of training classifiers on the post-probe period and time-generalizing them to the retention period only provides a limited access (hence a proxy) to the strength of memorized information, we would hesitate to interpret the results in *absolute* terms. When contrasting the conditions in relative terms, we find that anticipation of a strong distractor went along with relatively weak memorized information prior to distractor onset.

Interestingly, alpha/beta power prior to the presentation of the strong distractor decreased, whereas it was relatively sustained in the weak distractor condition. This effect is at odds with findings reported in the visual modality using an analogous paradigm (*Bonnefond and Jensen, 2012*), where — in line with idea of an inhibitory role for alpha oscillations (*Jensen and Mazaheri, 2010*; *Klimesch et al., 2007*) — induced alpha/beta enhancements were seen prior to a strong distractor. As in our study, the induced power effect in the Bonnefond and Jensen study was broadband and thus included the beta frequency range (see Figure 2 in *Bonnefond and Jensen, 2012*), although these authors focused only on the alpha (8–12 Hz) parts in their follow-up analysis. This, in general, is in line with our induced power effect shown in *Figure 3*, which shows peak effects around 13 Hz and 22 Hz. Our follow-up analysis, however, shows that both alpha and beta powers are relevant in a seemingly differential manner with respect to prioritization of memorized information. In the Bonnefond and Jensen study, pronounced pre-distractor evoked alpha power effects were observed that could not be identified in our auditory task. This negative finding could be modality specific but does also fit well with recent reported problems in identifying attentional pre-stimulus alpha-phase adjustment (*van Diepen et al., 2015*).

An integration of our alpha/beta findings with these lines of evidence appears challenging on the basis of power effects alone, as they could be interpreted either as the involuntary direction of attentional resources to the anticipated strong distractor (a sort of 'failed inhibition' within a gating account) or as a top-down driven prioritization of memorized information in anticipation of a strong distractor (see, for example, *Hanslmayr et al., 2016* and *van Ede, 2018*). The group-level effect of reduced decoding accuracy prior to the strong distractor could be seen to support the first interpretation, but this analysis does not relate the pre-distractor alpha/beta power to estimated strength of memorized information. A desirable analysis would be to perform a single-trial analysis within each participant, but obtaining a reliable quantification of strength of memorized information at a single-trial level is challenging (because of low signal-to-noise, for example). As an alternative approach, we sorted trials according to the pre-distractor power in the respective bands and performed a

median split. A prioritization account would entail two predictions: 1) that the low-power bin should be associated with relatively enhanced strength of memorized information prior to the distractor onset; and 2) that participants with more extreme power differences between the low- and high-power bins should also show more extreme differences in the strength of memorized information. We observed that for beta, the first prediction was met, whereas for alpha, it was the second prediction. Overall, while desynchronization in both frequency ranges appears to prioritize representations as suggested by other frameworks (*Hanslmayr et al., 2016*; *van Ede, 2018*), our results suggest functionally distinct contributions by auditory cortical alpha and beta oscillations. Interestingly, the extent to which alpha power was modulated within participants was significantly stronger than that for beta. Further studies will be needed to elaborate whether these distinct neural response patterns are linked to different cognitive processes that may subserve prioritization, such as temporal anticipation or selective attention. Altogether, our study underlines the value of combining conventional spectral analysis approaches with multi-voxel pattern analysis (MVPA) in advancing our understanding of the functional role of brain oscillations in the auditory system.

Our results may seem at odds with findings that frequently point to alpha power enhancements as an adaptive process within challenging listening tasks (for review see *Strauß et al., 2014*). In line with dominant views regarding the functional relevance of alpha oscillations (*Jensen and Mazaheri, 2010*; *Klimesch et al., 2007*), alpha enhancements in such circumstances have been linked to the selective inhibition of irrelevant 'channels' of auditory information (*Strauß et al., 2014*). In terms of tasks, a large proportion of studies illustrating enhanced task-related alpha power are not fully comparable to the present study because they are based on manipulation of selective attention (*Fu et al., 2001*; *Worden et al., 2000*) or on the processing of degraded (and thus more challenging) speech (but see alternative studies showing alpha decreases to track increased speech degradation, for example *McMahon et al., 2016* and *Miles et al., 2017*; see also *Hauswald et al., 2019*). Some studies also reported the functional relevance of alpha enhancements during retention periods of auditory working memory tasks (e.g. *Obleser et al., 2012*; *Wilsch et al., 2015*), but these studies did not introduce strong/weak distractors at predictable time points. Importantly, however, the sources of these alpha enhancement effects have most frequently been identified in non-auditory brain regions (e.g. *Obleser and Weisz, 2012*; *Wilsch et al., 2015*) and rarely in the auditory cortex (e.g. *Müller and Weisz, 2012*). Indeed, reduction of auditory cortical alpha activity has been commonly linked to attended (*Frey et al., 2014*) as well as perceived (including illusory) auditory input (e.g. *Leske et al., 2014*; *Müller et al., 2013*; *Weisz et al., 2007*; *Billig et al., 2019*; for a more general perspective see *Lange et al., 2014*). The association of alpha modulations to attended/ignored or perceived auditory information has to date been very indirect.

Our study significantly advances this state by showing a relationship between alpha/beta power in the left auditory cortex and strength of memorized information in the retention period. This result supports the interpretations of studies showing alpha power reductions in the auditory modality and a more general assertion that cortical alpha/beta desynchronization during memory tasks represents the content of memorized information (*Hanslmayr et al., 2016*). Similar to a recent fMRI study using a representational similarity approach (*Griffiths et al., 2019*), we show that suppression of alpha/beta power itself does not carry the information content but is likely to be a process that enables this to occur. In our study, which used broadband signals in a first step of the decoding analysis, this content-specific information appears to be largely driven by slow (delta/theta) activity, with the temporal fine structure containing relevant information on top of the slower amplitude changes. Overall, our results can be reconciled with those of previous studies focusing on alpha enhancements in auditory tasks, as alpha reductions or enhancements may show engagements of different neural systems in processing relevant or blocking irrelevant information, respectively (*van Ede, 2018*). The functional versatility of alpha power modulations in listening tasks also serves as a precaution not to equate, for example, alpha power enhancements simplistically to concepts such as listening effort (*McGarrigle et al., 2014*; *Pichora-Fuller et al., 2016*).

In summary, precise predictability of the occurrence of an auditory distractor leads to an anticipatory prioritization of memorized information. We show that modulations of alpha/beta oscillations in task-relevant auditory cortical regions could be a relevant process mediating the 'protection' of relevant auditory information against interference. In doing so, our study significantly adds to our understanding of the functional role of alpha and beta oscillations in the auditory system. Interestingly, our results suggest a somewhat differential and to date unreported pattern for these frequency

ranges in the auditory system: desynchronized beta states in task-relevant auditory regions appear to accompany the prioritization of information, but the extent of prioritization seems more level dependent for alpha. Illustrating a link to prioritization processes is a crucial first step in understanding the functional role of auditory cortical alpha/beta modulations during retention periods. Prospective studies using, for example, experimental neuromodulation techniques are needed to go beyond the correlational level. Future studies will need to test the extent to which the main direction of our findings generalizes to more natural listening situations such as speech, in which the temporal features of the distractor are not predictable with absolute precision.

## Materials and methods

### Participants

Thirty-three participants were included in the calculations (22 female; age range, 18–46 years; mean age, 26.8 years). Four participants were excluded because of technical issues during the testing or because the data were too noisy. All participants reported normal or corrected-to-normal vision and an absence of hearing problems in daily life. None of them suffered or was suffering a psychological or neurological disorder. Written informed consent was obtained from each participant before the experiment. They obtained either €10/h reimbursement or credits required for their bachelor studies in psychology. All procedures were approved by the Ethics Committee of the University of Salzburg.

### Stimuli and procedure

Participants underwent standard preparation procedures for MEG experiments. Five head-position indicators (HPI) coils were applied (three on the forehead, and one behind each ear). Using a Polhemus FASTRAK digitizer, anatomical landmarks (nasion, left and right pre-auricular points) and HPI coils were recorded, and additionally approximately 300 head-shape points were sampled. To control for eye movements and heart rate, electrodes were applied horizontally and vertically to the eyes (electrooculogram), one electrode was placed on the lower left ribs and one next to the right clavicle (electrocardiogram), as well as one reference electrode on the back. After entering the MEG cabin, a 5 min resting state was recorded, which was not utilized for the present study. The experimental paradigm consisted of a Sternberg task, similar to the one used by *Bonnefond and Jensen, 2012*, but adapted to the auditory modality. Visual stimuli were displayed with the PROPixx projector (VPixx Technologies Inc) on an opaque screen. Auditory stimuli were delivered using the SOUNDPixx system (VPixx Technologies Inc) through two pneumatic tubes. The stimulus delay introduced by the tube was measured using a microphone (16.5 ms ± 0.1 ms), and this delay was taken into account and compensated for in the analysis phase. The experiment was programmed in MATLAB 9.1 (The MathWorks, Natick, Massachusetts, USA) using the open source Psychophysics Toolbox (*Kleiner et al., 2007*).

During the experiment, participants focused on a fixation point. They listened to a memory set of four consonants spoken by a female voice (see *Figure 1A*). The interstimulus interval between the presentations of the consonants presentation was set to 1 s, and a distractor was presented to the participants 2 s after the final (fourth) letter (timepoint 1 s in *Figure 1A*). Within each experimental block, the distractor was either a consonant spoken by a male voice (strong distractor) or a temporally scrambled consonant (weak distractor). The scrambling of the distractor was achieved using the Matlab-based *shufflewins* function (*Ellis, 2011*). This scrambling approach preserves the frequency content of the original voice but makes it unintelligible. One second after the distractor, the probe was presented spoken by the same female voice as in the memory set. Thereafter, the participants needed to decide via button press whether the probe was part of the memory set or not. The participants were exposed to 12 blocks, six per each distractor condition and each one containing 24 trials. An intertrial interval from 1.5 to 2.5 s (mean 2.0 s, uniformly distributed) was used. One block had a duration of about 6 min. The sequence of the conditions and the assignment of the buttons was randomized across participants.

### MEG acquisition and analysis

The brain magnetic signal was recorded (sampling rate, 1 kHz; hardware filters, 0.1–330 Hz) using a whole-head MEG device (Elekta Neuromag Triux, Elekta Oy, Finland) in a standard passive

magnetically shielded room (AK3b, Vacuumschmelze, Germany). Signals were captured by 102 magnetometers and 204 orthogonally placed planar gradiometers at 102 different positions. We used a signal space separation algorithm (*Taulu et al., 2005*) implemented in the Maxfilter program (version 2.2.15) provided by the MEG manufacturer to remove external noise from the MEG signal (mainly 16.6 Hz, that is Austrian train AC power supply frequency, and 50 Hz plus harmonics) and to realign data to a common standard head position (to [0 0 40] mm, *-trans default* Maxfilter parameter) across different blocks on the basis of the measured head position at the beginning of each block.

First, a high-pass filter at 0.5 Hz (6th order zero-phase Butterworth filter) was applied to the continuous data. Then, continuous data were epoched around the onset of the retention phase using a 3-s pre- and post-stimulus window. For most analyses, the data were downsampled to 256 Hz (100 Hz to speed up decoding analysis, described below). The epoched data were subjected to an independent component analysis (ICA) using the runica algorithm (*Delorme and Makeig, 2004*). The Maxwell filtering greatly reduced the dimensionality of the data, from the original 306 sensors to usually 55–75 real components, depending on the single data block (*Elekta Neuromag MaxFilter User's Guide, 2012*). Therefore, prior to the ICA computing stage, a principal components analysis (PCA) with a fixed number of components (n = 50) was performed in order to ease the convergence of the ICA algorithm (see *Demarchi et al., 2019*, for example). The ICA components were manually scrutinized to identify eye blinks and eye movements and to train artifact and heartbeat, resulting in approximately two to five components that were removed from the data. The final rank of the data then ranged from 45 to 48. Given this extensive preprocessing, no trials had to be rejected.

In a first step, before time- and condition-generalizing trained classifiers, we applied an LDA classifier to a time window −0.2 s to 0.7 s around the probe presentation to confirm that by using all MEG sensors and all 288 trials we could decode whether a probe was part of the four-item memory set or not (*Figure 2A*). For this purpose, we employed the standard settings of the MVPA toolbox, that is a five-fold cross-validation scheme (training on 230 trials, testing on 58 trial), stratified, repeated five times, averaging the AUC values across folds splits and repetitions to obtain a time-course of decoding performance (*Treder, 2018*). Apart from serving as a sanity check, this analysis also yielded training-time ranges of interest for the subsequent cross-condition (time-generalized) decoding analysis. Note that for this latter analysis, no cross-validation was performed anymore. The classifier weights from the temporal decoding analysis were used to identify areas containing informative activity (see below). For analysis (*King and Dehaene, 2014*) in which classifiers were trained on post-probe onset periods and time- and condition-generalized to the retention period separately for the strong and weak distractor conditions, we followed the following rationale: given that neural activity driven by a memorized probe should share features with the activities elicited by the letters in the memory set, for our purposes, this time-generalization step yields a quantitative proxy for the strength of memorized information. By focusing in particular on the 0.5-s period prior to the presentation of the distractor and a training time period (of 0.4–0.7 s) in which the classifier showed above-chance performance, we could test the extent to which the strength of memorized information was modulated in anticipation of the distracting sound.

Given the nature of our research question outlined in the introduction, we wanted to analyze pre-distractor alpha power modulations in task-relevant brain regions. For this purpose, covariance-corrected (*Haufe et al., 2014*) classifier weights were projected to source space using an approach adapted from *Marti and Dehaene, 2017*. A realistically shaped single-shell head model (*Nolte, 2003*) was computed by warping a template MNI brain to the participant's head shape. A grid with 1 cm resolution on the template brain was morphed to fit the individual brain volume and lead fields were computed for each grid point. This information was used along with the covariance matrix of all sensors computed via the entire 30-Hz low-pass filtered epoch to obtain LCMV spatial filters (*Van Veen et al., 1997*). These beamformer filters were subsequently multiplied with the aforementioned covariance-corrected classifier weights to obtain 'informative activity' (*Marti and Dehaene, 2017*) in source space (taking the absolute value on source level). In order to make this data more interpretable, we implemented a permutation approach converting these time series to z-values and testing them across participants against 0 (see below). Overall, this data-driven approach yielded meaningful neuroanatomical regions that differentiated whether a probe was part of the memory set or not. Given our particular interest in auditory processes, we focused on the lSTG, which was the region providing the most prominent informative activity. For this region, we used the beamformer filters to project the single trial data onto a lSTG virtual sensor (location of

peak effect in auditory cortex across participant) and applied spectral analysis to it. More precisely, we used Fourier transform of Hanning-tapered data applied to a frequency range of 2–30 Hz (in 1 Hz steps) and time shifted between a period of −1.5 s to 0.5 s around onset of the distractor (shifted in steps of 0.025 s). The time window for the spectral analysis was adapted to each frequency (four cycles) and the analysis was performed separately for the strong and weak distractor condition.

Data preprocessing, spectral and source analysis was done using the Fieldtrip toolbox (*Oostenveld et al., 2011*). For the decoding analysis, we used the Matlab-based open-source MVPA-Light toolbox (https://github.com/treder/MVPA-Light; *Treder, 2019*).

## Statistical analysis

The behavioral impact of the distractor types was tested using a paired t-test, comparing accuracies and reaction times. Given the hypothesis that the strong distractor would be detrimental to performance, one-tailed testing was performed. With regard to our trained classifier, decoding accuracy was tested against chance level (AUC = 0.5) between −0.2 s and 0.7 s around probe onset using a t-test. In order to make the source projected classifier weights ('informative activity') more interpretable, we generated randomly shuffled trial labels and re-ran the same classifier and source projection approach. This was done 500 times, and the empirically observed values at each time and grid point were z-transformed using the mean and standard-deviation from the randomized data. The z-transformed data were tested against 0 across participants using a t-test. Also, the time-generalized decoding analysis and the spectral analysis described above were assessed using a t-test comparing the strong and weak distractor conditions. To control for multiple comparisons, we employed a non-parametric cluster permutation approach as proposed by *Maris and Oostenveld, 2007* normally using 5000 randomizations. Finally, testing the relationship between power modulations and strength of memorized information was done in a two-fold manner separately for alpha (~13 Hz) and beta (~21 Hz) oscillations. Both approaches were based on the first approach tested, sorting trials into high- and low-power bins according to a 0.6-s pre-distractor time window of our lSTG region of interest. The trained classifier was then applied to a 0.5-s pre-distractor window for low- and high-power trials, and AUC values were averaged over a 0.4–0.7-s post-probe training time period (analogously to the approach shown in *Figure 2B*). A first test along the predictions shown in *Figure 1B* addressed the question of whether lower-power trials would go along with increased strength of memorized information. Results of the aforementioned analysis were first entered into a repeated measures ANOVA using frequency band (alpha, beta) and power (high, low) as factors. Planned contrasts (high vs low power) were followed up using a paired t-test. A second test along the predictions shown in *Figure 1B* addressed whether individuals showing more extreme modulations between low- and high-power trials would also show more extreme differences in the strength of memorized information between these bins. To this end, power modulations were operationalized using a $\log_{10}$([high power] / [low power]) ratio and correlated with the respective difference in the strength of memorized information. This analysis was performed separately for alpha and beta.

## Acknowledgements

We would like to thank Jens Gfroerer-Kötschau and Manfred Seifter for their support during data collection.

## Additional information

### Funding

The authors declare that there was no funding for this work.

### Author contributions

Nathan Weisz, Conceptualization, Data curation, Formal analysis, Supervision, Funding acquisition, Investigation, Visualization, Methodology, Writing - original draft, Project administration, Writing - review and editing; Nadine Gabriele Kraft, Data curation, Formal analysis, Investigation, Visualization, Writing - original draft; Gianpaolo Demarchi, Conceptualization, Data curation, Formal analysis, Investigation, Methodology, Writing - original draft, Writing - review and editing

### Author ORCIDs

Nathan Weisz (iD) https://orcid.org/0000-0001-7816-0037
Nadine Gabriele Kraft (iD) http://orcid.org/0000-0003-2818-2283
Gianpaolo Demarchi (iD) https://orcid.org/0000-0002-7597-9298

### Ethics

Human subjects: The study was conducted according to the declaration of Helsinki (7th revision). Written informed consent was obtained from each participant prior to the experiment. All procedures were approved by the Ethics Committee of the University of Salzburg (EK-GZ:22/2016a).

### Decision letter and Author response

Decision letter https://doi.org/10.7554/eLife.55508.sa1
Author response https://doi.org/10.7554/eLife.55508.sa2

## Additional files

### Supplementary files

• Transparent reporting form

### Data availability

All the preprocessed, downsampled raw data and processing MATLAB and R scripts could be found at https://doi.org/10.17605/OSF.IO/PW9RD.

The following dataset was generated:

| Author(s) | Year | Dataset title | Dataset URL | Database and Identifier |
|---|---|---|---|---|
| Weisz N, Kraft N, Demarchi G | 2020 | Auditory cortical alpha / beta desynchronization prioritizes the representation of memory items during a retention period | https://doi.org/10.17605/OSF.IO/PW9RD | Open Science Framework, 10.17605/OSF.IO/PW9RD |

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
