## [Decision Letter]

**Acceptance summary:**

The work describes a relationship between α activity and the prioritisation of elements held in working memory that will be of broad interest.

**Decision letter after peer review:**

Thank you for submitting your work entitled "Auditory cortical α desynchronization prioritizes the representation of memory items during a retention period" for consideration by *eLife*. Your article has been reviewed by three peer reviewers, one of whom is a member of our Board of Reviewing Editors, and the evaluation has been overseen by a Senior Editor. The following individual involved in review of your submission has agreed to reveal their identity: William Sedley (Reviewer #2).

Our decision has been reached after consultation between the reviewers. Based on these discussions and the individual reviews below, we regret to inform you that your work is not currently suitable for publication in *eLife*. It is *eLife* policy that if needed revisions would take more than 8 weeks, we reject a manuscript. In your case, that is where we stand: your paper is of potential interest, but requires significant revisions that we believe require more than 2 months to resolve.

Specifically, the reveiewers found the work interesting in measuring α changes in auditory areas measured using MEG as a function of the presence of a (temporally predictable) auditory distractor. The authors use a modified Sternberg technique with strong and weak distractors presented in different blocks and a phonological WM task after Bonnefond and Jensen's visual version. The group data show decreases in pre-distractor induced but not phase locked α power in left temporal cortex and lower decoding accuracy for the target letter before the strong distractor compared to the weak. In contrast, analysis of the individual α changes showed that increased α correlated with decreased decoding accuracy at the time of the distractor explaining about 1/3 of variance.

We encourage you to consider resubmitting the paper to *eLife* if you are able to address the issues described below. The most significant points include potential circularity in the analysis, the specificity of the effect with respect to α, the specification of time windows, and the discussion of the basis.

Major comments:

1) The cover mentions 'In daily life we are forced to prioritize ongoing sensory information, to be able to deal with the tremendous amount of input we are exposed to.' Does a situation ever occur in daily life when we have to deal with a distractor that is entirely predictable in time?

2) The central message of this paper relates to α, but the data suggests it is more of a β effect (e.g. the difference plot in Figure 3A, the 10-16Hz band used for the correlation analysis). While the authors make a strong case in the Introduction for the effect to arise in the α band, the data disputes this. Yet, the authors continue to focus on α throughout the Discussion.

3) Throughout the analysis, the authors use restricted windows of analysis. Such practices help limit the multiple comparison problem, but strong a priori reasons are required for window selection – reasons that are not supplied in the manuscript. Were these windows were selected post-hoc (intentionally or not)? This is most troublesome at the start of subsection “Pre-distractor α power modulations of probe-related information”, where the α band was defined as 10-16Hz. This selection is particularly worrying as it does not appear to be either a priori (10-16Hz is really pushing the boundaries of what is consider α) or driven by earlier analysis (the effect in Figure 3A extends from 10-25Hz). Given that this effect is only marginally significant (p = 0.0272), the concern is that worry these parameters may have been selected to drop the p-value below the "significance" threshold, and would not survive multiple comparison correction should the window be expanded.

4) The reviewers struggled with the rationale of the decoding approach. Subsection “Decoding probe-related information” paragraph one describes beginning by decoding from post-probe activity whether that probe was part of the memory set or not. Is decoding accuracy here intended to act as a proxy for how well the memory set was encoded/retained in memory? If so, shouldn't that accuracy be greater than chance earlier than 300 ms after the probe onset (Figure 2A shows it is at chance until then)? This also seems rather an indirect proxy – presumably the neural activity post-probe also heavily features the identity of the probe itself. Furthermore, the neural activity involved in distinguishing presence vs. absence in the memory set might also reflect differential adaptation of the probe representation (although the timescales are probably long enough for this not to be a consideration in sensory cortices). Perhaps decoding accuracy is not supposed to be a proxy for how well the memory set was encoded/retained in memory, since later (paragraph two and elsewhere) the authors refer to using this trained classifier to look for how well the probe is represented during the retention period. To address this question we might have expected the authors to have trained a classifier on neural representations of specific letters (e.g. using the period after each is presented during the memory set, or when they appear as probes) and test for these representations during the retention period. With such an analysis the expectation might be that in a successful trial (and perhaps more so for a weak distractor), that trial's probe letter would be better neurally represented during retention if it were present in the memory set but not if it were absent. We may have misunderstood the rationale of the analyses, in which case it likely needs to be set out more clearly in the manuscript.

5) A major concern was the potential circularity of the analysis. The same signal was used to get the classifier estimates and the α/β power. If α/β contributes to the classifier then the signal would essentially be correlated with itself. There are ways to get around this issue, for instance by using a band-stop filter (i.e. taking out α/β from the signal) on the data before feeding it into the classifier. The reviewers suggest orthogonalizing their α/β data and classifier data before attempting a correlation.

6) We would like to see the brain distribution of the AUC in Figure 2A as a function of time. We cannot work out what latency the brain data shown correspond to. We agree the data are left lateralised but there seems to be an effect in right operculum and DLPFC.

7) From 2B left and right, the decoding of probe-related information prior to the distractor seems to be at/below chance for strong distractors and below chance for weak distractors (assuming chance is AUC = 50%). On that basis, I'm not sure whether the difference (shown in 2B right) underpinning one of the paper's main conclusions is really interpretable. The authors do not mention or discuss this.

[Editors’ note: further revisions were suggested prior to acceptance, as described below.]

Thank you for resubmitting your work entitled "Auditory cortical α desynchronization prioritizes the representation of memory items during a retention period" for further consideration by *eLife*. Your revised article has been evaluated by Barbara Shinn-Cunningham (Senior Editor) and a Reviewing Editor.

The authors have addressed a number of concerns raised in a previous round, importantly providing a clearer rationale for their approach, and checking the α-specificity of the effects. The additional analysis of memory content decoding by frequency band has also strengthened the paper. There are two remaining issues that need to be addressed before acceptance, as outlined below.

Major Issue 1: Below chance classifier decoding.

1) There does not seem to be any reference to the cross-validation of the classifier in the main text. Before seeing whether the classifier can generalise to the retention window, it would be important to see whether the classifier can generalise across different folds of the probe window. This supplementary analysis would allow more confidence in the validity of the classifier.

2) A lack of detail about the classification approach in the Materials and methods makes it difficult to evaluate the suitability of said classification approach. It would be helpful to note explicitly the number of trials used for training/testing, and the number and nature of features included.

3) The below chance decoding in paragraph two of subsection “Decoding probe-related information” is a major concern, particularly because of the apparent absence of cross-validation during the probe window. It would be important to demonstrate that:

i) This is not due to overfitting to the training dataset. It is unclear from the Materials and methods, but perhaps 306 sensors and 50 timepoints (i.e. 15,300 features) have been used to train on (by my count) 288 trials. Such a large number of features relative to a low number of trials can mean that the classifier cannot generalise to any data set other than the training data set (potentially explaining the below chance decoding).

ii) This is not due to linear dependence between features. Again, it is unclear, but it would appear that raw time/sensor data has been used as features in the classifier. Though MEG has excellent spatial resolution, there is still some co-linearity in the signal between sensors. This can impede the generalisability of the classifier to test data sets.

Both of these points could be addressed by running a PCA on the features prior to classification. PCA will help minimise linear dependencies between features, and if one then takes only the top 100 components, then the number of features will not exceed the number of training trials.

Major Issue 2: Correlation between α power variability and ability to decode memorandum.

The authors have correlated, across subjects, a measure of α power variability with the effect of α power on memorandum decodability. As a previous reviewer points out, this is rather removed from the data and does not clearly support the interpretation that α power and decodability themselves are linked. A trial-by-trial analysis of α power versus memorandum decodability was suggested. This would require some consideration of across-subject differences in the raw measures. So either a correlation could be performed subject by subject, and then the correlation coefficients tested against zero as a group, or a single trial-by-trial regression could be run across subjects with appropriate normalization or inclusion of random subject effects.

It might be that the trial-wise data are too noisy for such an approach, and that this is a reason for the authors' use of the median split into high-α and low-α trials. However, the appropriate analysis would then seem to be to compare memorandum decodability for low-α versus (minus) high-α trials, testing whether this difference is significantly greater than zero for the group. The correlation shows only that α power variability is associated with the effect of α power on memorandum decodability. This might not be meaningless, but as the scatter plots in 4B show, there is a fairly balanced split of subjects for whom higher α is associated with better decoding and those for whom lower α is associated with better decoding (positive and negative differences on the y axis). The axis labels are not too clear as to the direction of the comparisons. A possible interpretation is that for subjects on the left of the scatter plots with a lower log10 α ratio (nearer 0.7, i.e. nearer 10^0.7 = four times more α power in high- than low-α trials), high-α trials lead to better decoding than low-α trials, whereas for subjects on the right of the scatter plots with a higher log10 α ratio (nearer 0.9, i.e. nearer 10^0.9 = eight times more α power in high- than low-α trials), the reverse is true.

In the absence of clarification or a new analysis along the lines suggested, it is not clear that the data support the conclusions of the paper.

---

## [Author Response]

Major comments:1) The cover mentions 'In daily life we are forced to prioritize ongoing sensory information, to be able to deal with the tremendous amount of input we are exposed to.' Does a situation ever occur in daily life when we have to deal with a distractor that is entirely predictable in time?

We do not claim that our task is in particular ecologically valid and agree that such situations of perfect predictable time are very rare, albeit possible (e.g. ticking of clock, dripping of faucet, also listening to an “overlearned” song).

Nevertheless, when planning the experiment we decided that sticking to the original visual version of the experiment by Bonnefond and Jensen would make a lot of sense. In particular, we were hoping that maximising the similarity to their study would boost chances to find a pre-distractor increase of phase consistency (as would be seen in the evoked response). As described in the manuscript we were unable to identify such an effect in the auditory modality, which however fits well with general problems in finding attentionally driven α phase effects in (stimulus-free) prestimulus periods (e.g. van Diepen et al., 2015). Overall, this domain appears to be quite controversial (van Diepen et al., 2019) and in the current version we are not placing excessive emphasis on this controversy.

In order to acknowledge this issue we added some additional comments to the Discussion.

“While perfect temporal predictability of a distractor is rare in natural environments (e.g. ticking of clock, dripping faucet), this was maintained in the present study to assure maximum comparability. In particular strict particular temporal predictability should boost some potential effects, especially ones pertaining to increasing pre-distractor phase consistency.”

“Future studies will need to test to what extent the main direction of our findings generalize to more natural listening situations such as speech in which temporal features of the distractor are not predictable with absolute precision.”

2) The central message of this paper relates to α, but the data suggests it is more of a β effect (e.g. the difference plot in Figure 3A, the 10-16Hz band used for the correlation analysis). While the authors make a strong case in the Introduction for the effect to arise in the α band, the data disputes this. Yet, the authors continue to focus on α throughout the Discussion.

The response to this very justified issue should be split into two parts: one pertaining to the somewhat appearing arbitrariness of the chosen time- and frequency windows especially for the correlation analysis and another pertaining the labels for frequency boundaries.

Regarding the first issue we apologize that the impression of arbitrariness was elicited. However the selection of time and frequency windows (following a reasonable restriction of the “search-window”; see Author response image 1) was strictly data driven according the spectrotemporal characteristics of the negative cluster resulting from the induced power contrast (shown in Figure 3A of the manuscript). Relevant features are displayed in the Author response image 1.

**Author response image 1. sa2fig1:** The image on the left shows the (unthresholded) T-values averaged between in the. 5 to 1s period, i.e. the.5s window preceding the distractor. Separate (negative) peaks can be identified at 13 Hz and 22 Hz. Also when inspecting which frequencies contribute to the negative cluster (middle panel) it is clear that the cluster comprises two peaks at the respective frequencies. Displaying the temporal profiles for these frequencies (right panel) it can be seen that peaks are reached at ~.7s, i.e. ~.3s prior to the presentation of the distractor. Given these follow-up inspections of the cluster, in the original manuscript we focused on the lower-frequency contribution, i.e. multitaper FFT centred at 13Hz and.7s using reasonable parameters to estimate power.

We apologize for this lacking detail and added clarifications:

“To address the functional relevance of pre-distractor α power modulations in the left STG in greater detail, trials were sorted according to α power (13 +/- 3 Hz) in this region in a 400-1000 ms time period following the onset of the retention period (i.e. a 600 ms pre-distractor window centered on peak latency effect at 700 ms).”

It may be argued whether effects at 13 Hz -i.e. not strictly falling into the canonical Berger-band- can be labeled as “α” or not. But the reviewers definitely are right in noting that we ignored the higher frequency (~22 Hz) part in the description as well as in the follow-up analysis. In fact upon re-inspecting the screenshot, it appears possible that the “true” effect is broadband but that noise at 16.6 Hz (stemming from the railway lines in the vicinity of the lab; perhaps not fully removed despite using maxfilter to clean the data as well as analysing data in source space) obscure it. This however cannot be resolved conclusively. In general, a broadband effect is in line with more current developments that see α and β describing similar processes: e.g. a recent paper in *eLife* (Griffiths et al., 2019) simply used the shorthand “α / β” label. Also upon reinspecting the study by Bonnefond and Jensen, which served as a model for ours, it is quite clear when inspecting their crucial Figure 2, that their pre-distractor effects are also rather broadband ranging from ~8-20 Hz. It is just that the authors decided to restrict their follow-up analysis to the canonical Berger-band (8-12 Hz; as indicated by the box). We raise awareness of the broadband nature of the induced power effect in Figure 3 at different points, e.g.:

“This impression is supported by a nonparametric permutation test (*p* cluster = .0104), yielding a significant difference in this period with peak differences ~12-13 Hz and ~21-22 Hz (Figure 3A, right panel) comparable to the study in the visual domain (Bonnefond and Jensen, 2012). ”

“Furthermore, given the broader spectral distribution of the power effect (Figure 3A) , we reran the entire analysis described in this section using β power (centred at peak effect at 22 +/ 3 Hz) to sort trials. No negative cluster was obtained when decoding memorized information (one positive cluster at *p* = .67), meaning that at no time point did a correlation coefficient become statistically significant even at an uncorrected level (*r*’s at.895 s and 1.304 s = .06 and.19 respectively).”

“As in our study, the induced power effect in the Bonnefond and Jensen study was broadband including the β frequency range (see Figure 2 in (Bonnefond and Jensen, 2012)) , however the authors only focused on the α (8-12 Hz) parts in their follow-up analysis. This in general is in line with our induced power effect shown in Figure 3, which shows peak effects around 13 Hz and 22 Hz. Our follow-up analysis however show that only the ~13 Hz (“α”) part is relevant with respect to prioritization of memorized information.”

Despite the induced power effect being broadband however, it appears that functionally -i.e. with respect to our inhibition vs prioritization question- not all frequency parts are equal within this band. In particular we reran the interindividual correlation analysis and now show that the prioritization effect seen is specific to the ~13 Hz part. Given this important follow-up analysis inspired by the reviews and the new accompanying explanations is the text, we hope that you are fine in maintaining the “α” label for the ~13 Hz effect.

3) Throughout the analysis, the authors use restricted windows of analysis. Such practices help limit the multiple comparison problem, but strong a priori reasons are required for window selection – reasons that are not supplied in the manuscript. Were these windows were selected post-hoc (intentionally or not)? This is most troublesome at the start of subsection “Pre-distractor α power modulations of probe-related information”, where the α band was defined as 10-16Hz. This selection is particularly worrying as it does not appear to be either a priori (10-16Hz is really pushing the boundaries of what is consider α) or driven by earlier analysis (the effect in Figure 3A extends from 10-25Hz). Given that this effect is only marginally significant (p = 0.0272), the concern is that worry these parameters may have been selected to drop the p-value below the "significance" threshold, and would not survive multiple comparison correction should the window be expanded.

Again we apologize that choices appear arbitrary, even though they are strictly literature- and data-driven. The time- and frequency-window used in our study almost perfectly overlap with the one used in the Bonnefond and Jensen study (5-30 Hz with focus of statistical analysis on.5 s prior to distractor onset; showing main effect spread between 8-20 Hz; see above). We identified the strongest induced power effects to be ~.7 s and at ~13 and ~22 Hz (see Author response image 1). We admittedly ignored the higher frequency (~22 Hz) part in our original submission as rightly pointed out by the reviewers. A window of +/-.3 s (centred on.7 s) is quite a standard choice for estimating power via FFT, thus the described.4 to 1 s time window for the correlational analysis between power changes and decoding. In the current version of the manuscript we used the same time-window to estimate the correlation effects with the β power changes.

While the time- and frequency-bands were data-driven from the induced power analysis, we did not average over time for the analysis correlating the power with the decoding changes. Thus the negative clusters are also purely driven by the data (scatter plots provided only for visualization, making sure that effects are “convincing”, i.e. not outlier-driven). It may be argued whether to call a *p* = .027 “marginally significant” (similar to other variations on “significant” sometimes encountered in the literature), but we think that this pre-distractor effects should be appreciated together with the post-stimulus correlation (*p* = .001) overall supporting the claims forwarded by this analysis. It is challenging to follow-up (and thus not pursued further in the manuscript), but it is plausible that the “true” cluster may actually encompass the pre- and post-stimulus periods, but that any effect is transiently interrupted by the evoked effects influencing power as well as decoding results.

4) The reviewers struggled with the rationale of the decoding approach. Subsection “Decoding probe-related information” paragraph one describes beginning by decoding from post-probe activity whether that probe was part of the memory set or not. Is decoding accuracy here intended to act as a proxy for how well the memory set was encoded/retained in memory? If so, shouldn't that accuracy be greater than chance earlier than 300 ms after the probe onset (Figure 2A shows it is at chance until then)? This also seems rather an indirect proxy – presumably the neural activity post-probe also heavily features the identity of the probe itself. Furthermore, the neural activity involved in distinguishing presence vs. absence in the memory set might also reflect differential adaptation of the probe representation (although the timescales are probably long enough for this not to be a consideration in sensory cortices). Perhaps decoding accuracy is not supposed to be a proxy for how well the memory set was encoded/retained in memory, since later (paragraph two and elsewhere) the authors refer to using this trained classifier to look for how well the probe is represented during the retention period. To address this question we might have expected the authors to have trained a classifier on neural representations of specific letters (e.g. using the period after each is presented during the memory set, or when they appear as probes) and test for these representations during the retention period. With such an analysis the expectation might be that in a successful trial (and perhaps more so for a weak distractor), that trial's probe letter would be better neurally represented during retention if it were present in the memory set but not if it were absent. We may have misunderstood the rationale of the analyses, in which case it likely needs to be set out more clearly in the manuscript.

Thank you for making it very clear to us that the entire rationale remains obscure based on the original manuscript. Next to adding more explanations at strategically chosen points, two measures were undertaken to improve comprehensibility: 1) We changed Figure 1 to include a schematic depiction of the analysis rationale as well as the predictions based on the inhibition vs prioritization account. The -not very informative- bar plots of the behavioral effects have been removed (effects only described in text). 2) We tried to streamline the terminology by referring to the time-generalized decoding effects as (strength of) “memorized information”. Indeed our analysis approach of training the classifier to post-probe periods and time-generalizing into the retention period makes it (as rightly indicated in your comment) a “proxy” of memorized information. Given that a probe belonging to the memory set should share features in common with the actual memory set, we think that this argumentation is justified, especially when it comes to contrasting the experimental conditions.

Some additional information has been added to the text hopefully helping to make the approach more clear. e.g.: “The rationale for this approach (see also Figure 1) was that if a probe was part of the memory set, it should share neural patterns with those elicited by the actual memory set. This should not be the case for probes that were not part of the memory set, so that time-generalizing these post-probe patterns to the retention period should give a quantitative proxy for the fidelity of the memory representation.”

Regarding the latency of the probe-locked effects, our results show that early latencies do not contain sufficient information on whether an item was part of the memory set or not, meaning that at a low (e.g. early sensory) level similar processes are involved. While we did not set out to identify such late effects (i.e. they are again data-driven), this finding fits actually very nicely with other reports. For example EEG studies on old vs new memory effects consistently report latencies beyond 300 ms.

We added following sentence: “The time course of this effect is very much in line with evoked response studies on old vs new effects in short term memory (Danker et al., 2008; Kayser et al., 2003), indicating that early sensory activation is not informative on whether a probe was a memorized item or not.”

5) A major concern was the potential circularity of the analysis. The same signal was used to get the classifier estimates and the α/β power. If α/β contributes to the classifier then the signal would essentially be correlated with itself. There are ways to get around this issue, for instance by using a band-stop filter (i.e. taking out α/β from the signal) on the data before feeding it into the classifier. The reviewers suggest orthogonalizing their α/β data and classifier data before attempting a correlation.

This is a very interesting issue that we overlooked in our initial approach, even though the statement that “The same signal was used to get the classifier estimates and the α/β power” is not quite correct: the classifiers were trained post-probe presentation whereas α / β was estimated during the cue-target interval. The fact of time-generalizing the classifier (trained on post-probe neural activity) to the retention period should actually greatly diminish the danger of “circularity”. Nevertheless it is correct that if the post-probe decoding results are driven mainly by α (power) modulations and strong α differences are observed prior to the distractor, then some ambiguity exists. We followed-up on this issue in a dual manner:

Firstly we performed the post-probe decoding analysis using data filtered in different bands (broadband, theta, α and β) and either keeping or abolishing the temporal fine-structure (using the norm of the hilbert transform). These results described in detail in a new section (subsection “Memory-related information is mainly carried by low-frequency activity”) and Figure 5 clearly shows that neither α or β activity contribute significantly to the decoding. This seems to be mainly driven by slower frequencies, with the temporal fine structure adding relevant information on top of the amplitude changes.

Secondly, we repeated the correlation analysis presented in Figure 4 also using pre-distractor β power (~22 Hz) to sort the trials and are able to confirm that the relationship to the decoding of memorized information is specific for α.

Altogether this shows that α power reductions to do not carry representational information, but -in a sense of a prioritization signal- enables such content specific patterns to emerge, a conclusion much in line with a recent *eLife* paper (Griffiths et al., 2019).

We added a statement:

“This result supports the aforementioned interpretations of studies showing α power reductions in the auditory modality and a more general assertion of cortical α desynchronization during memory tasks to support representing the content of memorized information (Hanslmayr et al., 2016). Similar to a recent fMRI study using a representational similarity approach (Griffiths et al., 2019), we show that suppression of α power itself does not carry the information content but likely is an enabling process for this to occur. In our study, in which we used broadband signals in a first step of the decoding analysis, this content-specific information appears to be largely driven by slow (δ / theta) activity with the temporal fine structure containing relevant information on top of the slower amplitude changes.”

6) We would like to see the brain distribution of the AUC in Figure 2A as a function of time. We cannot work out what latency the brain data shown correspond to. We agree the data are left lateralised but there seems to be an effect in right operculum and DLPFC.

We have modified Figure 2A now to also include the temporal evolution of “informative activity” (i.e. source projection of classifier weights; the AUC is captured in the sensor level using all sensors). For the relevant time-window it can be clearly seen that strong informative activity emerges early between 300-500 ms following probe onset with a dominance in left STG and subsequently spreads to other regions (including parietal, DLPFC and inferior temporal regions). Given its reported importance for phonological short term memories (e.g. Jaquemot and Scott, 2006) and being considered as auditory processing region, we find that focusing our induced activity analysis on this area can be justified.

7) From 2B left and right, the decoding of probe-related information prior to the distractor seems to be at/below chance for strong distractors and below chance for weak distractors (assuming chance is AUC = 50%). On that basis, I'm not sure whether the difference (shown in 2B right) underpinning one of the paper's main conclusions is really interpretable. The authors do not mention or discuss this.

This is an excellent observation. Indeed it would be intuitive to expect constant activation of memorized information (i.e. reflected in electrophysiological recordings). However, in general we do not think that below chance decoding invalidates the main conclusions of the paper, as these rely on the contrast, either between weak vs strong distractors (Figure 3) or α power (Figure 4). Training of the classifier was done on post-probe periods and increases of decoding accuracy would appear the more when similar -putatively content-related- patterns occur in the retention period. In case this pattern is not activated (significant) below chance decoding accuracy would be the consequence.

In general it is evident from the time courses in Figure 3B that neural patterns related to memorized information ramp up slowly, being largely absent in periods prior to the distractor (which could appear as below chance decoding in the result) and becoming stronger towards anticipated onset of the probe item. Our results effectively show that the process of ramping up neural patterns related to memorized information is delayed when a strong distractor is expected. Activation in a timely “on-demand” fashion seems like an economic functional organization. Alternative forms of maintaining memorized information could be engaged in the pre-distractor period, that do not lead to striking surface-level electrophysiological patterns. The possibility of network-level “activity-silent” memory representations has been recently demonstrated suggested and demonstrated by Stokes et al. (Stokes et al., 2015; Wolff et al., 2017).

We address this issue now explicitly mentioned:

“In fact, when post-hoc testing the decoding accuracy over the entire aforementioned test- and training-time window, average decoding accuracy prior to the strong distractor was significantly below chance (*t*32 = -3.98, *p* = 3.63e-04) whereas it did not differ for weak distractors (*t*32 = -1.73, *p* = .09). Below-chance decoding may appear surprising but simply means a relative absence of memory-item specific patterns akin to those elicited during relevant periods following probe onset. Since this activation gradually ramps up toward the onset of the probe in both conditions albeit somewhat delayed in the strong distractor condition, it is clear that some representation with regards to the memorized item would also need to be present during the periods of below chance decoding. Recently Stokes et al. (Stokes, 2015; Wolff et al., 2017) described network-level “activity silent” processes encoding working memory content and similar processes could be present early during the retention period in the present task.”

[Editors’ note: further revisions were suggested prior to acceptance, as described below.]

Major Issue 1: Below chance classifier decoding.1) There does not seem to be any reference to the cross-validation of the classifier in the main text. Before seeing whether the classifier can generalise to the retention window, it would be important to see whether the classifier can generalise across different folds of the probe window. This supplementary analysis would allow more confidence in the validity of the classifier.

Thanks for pointing out this important issue, that we missed to describe in the previous version of the manuscript. A cross-validation approach was actually already implemented to test where out 9temporal) classifier(s) could decode whether a probe was part of the memory set or not (i.e. AUC values in Figure 2A are the average across the different folds). We have now added the missing details to the text. In essence, for the pure temporal decoding step shown in Figure 2A “standard” (i.e. common in the field) 5-fold repeated 5 times cross validation scheme was performed (see e.g. http://www.fieldtriptollbox.org/tutorial/mvpa_light/#cross-validation).

“In a first step, before time- 684 and condition-generalizing trained classifiers, we applied an LDA classifier to a time window -.2 to.7 s centered on the probe presentation to confirm that by using all MEG sensors and all 288 trials we could decode whether a probe was part of the four-item memory set or not (Figure 2A). For this purpose, we employed the standard settings of the MVPA toolbox, that is a 5-fold cross-validation scheme (training on 230 trials, testing on 58 trial), stratified, repeated 5 times, averaging the AUC values across folds splits and repetitions to obtain a time-course of decoding performance (Treder, 2018). Apart from serving as sanity check, this analysis also yielded training-time ranges of interest for the subsequent cross-condition (time-generalized) decoding analysis. Note that for this latter analysis no cross-validation was performed anymore. The classifier weights from the temporal decoding analysis were used to identify areas containing informative activity (see below). For analysis (King and Dehaene, 2014) in which classifiers were trained on post-probe onset periods and time- and condition generalized to the retention period separately for the strong and weak distractor condition we followed the rationale: Given that neural activity driven by a memorized probe should share features with that elicited by the ones in the memory set, for our purposes this time-generalization step yields a quantitative proxy for the strength of memorized information.”

2) A lack of detail about the classification approach in the Materials and methods makes it difficult to evaluate the suitability of said classification approach. It would be helpful to note explicitly the number of trials used for training/testing, and the number and nature of features included.

We apologize for not having provided these important details in the previous manuscript version. Relevant information about the classifier parameters have now been provided. (See response 1).

3) The below chance decoding in paragraph two of subsection “Decoding probe-related information” is a major concern, particularly because of the apparent absence of cross-validation during the probe window. It would be important to demonstrate that:i) This is not due to overfitting to the training dataset. It is unclear from the Materials and methods, but perhaps 306 sensors and 50 timepoints (i.e. 15,300 features) have been used to train on (by my count) 288 trials. Such a large number of features relative to a low number of trials can mean that the classifier cannot generalise to any data set other than the training data set (potentially explaining the below chance decoding).ii) This is not due to linear dependence between features. Again, it is unclear, but it would appear that raw time/sensor data has been used as features in the classifier. Though MEG has excellent spatial resolution, there is still some co-linearity in the signal between sensors. This can impede the generalisability of the classifier to test data sets.Both of these points could be addressed by running a PCA on the features prior to classification. PCA will help minimise linear dependencies between features, and if one then takes only the top 100 components, then the number of features will not exceed the number of training trials.

We would like to thank the Editors for giving us the opportunity to elaborate on this important issue. First of all we wanted to point out that for the sanity check in Figure 2A actually a quite standard cross validation approach was pursued (see response 1), however we missed communicating this important details. Nevertheless “below chance decoding” deserves more attention than we previously dedicated in the manuscript. Addressing this issues requires in a first step to exclude being a consequence of flawed analysis. In a second step, it requires some (at least attempted) clarifications what this could mean.

Regarding the potential of flawed analysis: we are confident that overfitting is not a severe issue in this study. There are different reasons for this, that would have been more obvious if we have provided the aforementioned requested details already in our previous version of the manuscript. 1) Our classifier(s) trained to decode whether a probed was part of a memory set or not (Figure 2A) were tested using a -quite standard- 5 times 5-fold cross validation approach, which reduces the peril of overfitting (see response 1). 2) a classifier was trained / tested for each time-point separately, which leaves us the 306 features per time-point. The use of Maxfilter for cleaning and repositioning the MEG data effectively reduces the number of independent “components” (ie rank) to 55-75, serving effectively as a first run of (PCA-like) feature selection. Moreover, at the stage of the ICA cleaning, a PCA with n=50 components is performed, to facilitate the convergence of the ICA algorithm. This has been now clarified in the text:

“The Maxwell filtering greatly reduces the dimensionality of the data, from the original 306 sensors to usually 55-75 real components, depending on the single data block (Elekta, 2012). Therefore, prior to the ICA computing stage, a PCA with a fixed number of components (n=50) is performed, in order to easen to convergence of the ICA algorithm (see e.g. Demarchi et al., 2019). The ICA components were manually scrutinized to identify eye blinks, eye movements, train artifact and heartbeat, resulting in approximately two to five components that were removed from the data. The final rank of the data then ranged from 45 to 48. Given this extensive preprocessing, no trials had to be rejected.”

“Interestingly, classifier accuracy especially prior to the strong distractor was significantly below chance level. Such patterns are not uncommon in M/EEG studies using time- and condition generalized decoding (see King and Dehaene, 2014) and are also seen in fMRI studies (e.g. van Loon et al., 2018). Descriptively, in electrophysiology below chance decoding can arise when neural patterns underlying representations are opposing and/or temporally shifted. Thus below chance decoding cannot be interpreted as absence of condition- or feature relevant information. However, a functional interpretation is challenging (King and Dehaene, 2014). Based on the fact that our approach training classifiers on the post-probe period and time-generalizing them to the retention period only provides a limited access (hence a proxy) to the strength of memorized information we would hesitate to interpret results in *absolute* terms. Contrasting the conditions in relative terms, we find that anticipation of a strong distractor went along with relatively weaker memorized information prior to distractor onset.”

Major Issue 2: Correlation between α power variability and ability to decode memorandum.The authors have correlated, across subjects, a measure of α power variability with the effect of α power on memorandum decodability. As a previous reviewer points out, this is rather removed from the data and does not clearly support the interpretation that α power and decodability themselves are linked. A trial-by-trial analysis of α power versus memorandum decodability was suggested. This would require some consideration of across-subject differences in the raw measures. So either a correlation could be performed subject by subject, and then the correlation coefficients tested against zero as a group, or a single trial-by-trial regression could be run across subjects with appropriate normalization or inclusion of random subject effects.It might be that the trial-wise data are too noisy for such an approach, and that this is a reason for the authors' use of the median split into high-α and low-α trials. However, the appropriate analysis would then seem to be to compare memorandum decodability for low-α versus (minus) high-α trials, testing whether this difference is significantly greater than zero for the group. The correlation shows only that α power variability is associated with the effect of α power on memorandum decodability. This might not be meaningless, but as the scatter plots in 4B show, there is a fairly balanced split of subjects for whom higher α is associated with better decoding and those for whom lower α is associated with better decoding (positive and negative differences on the y axis). The axis labels are not too clear as to the direction of the comparisons. A possible interpretation is that for subjects on the left of the scatter plots with a lower log10 α ratio (nearer 0.7, i.e. nearer 10^0.7 = four times more α power in high- than low-α trials), high-α trials lead to better decoding than low-α trials, whereas for subjects on the right of the scatter plots with a higher log10 α ratio (nearer 0.9, i.e. nearer 10^0.9 = eight times more α power in high- than low-α trials), the reverse is true.In the absence of clarification or a new analysis along the lines suggested, it is not clear that the data support the conclusions of the paper.

This is an insightful comment and after careful deliberation we agree that the correlation approach alone goes not full support the interpretation that desynchronized states in auditory cortex are related to prioritization of presentations in auditory working memory. We also agree that ideally power and classifications results could be meaningfully linked at a single trial level, but that this prone to fail given the noiseness of the data. We thus implemented an analysis along the lines as suggested by the reviewer comparing strength or memorized information (see above) between high and low power in the α and β band (i.e. matching the peaks effects shown in Figure 3A). A prioritization account would predict that lower power (i.e. desynchronized states) should exhibit greater strength of memorized information as compared to high power states (and vice versa for an inhibition account). We also maintained a correlation approach analogous to the previous submission and attempted to explain the rationale for this in clearer terms: i.e. more extreme differences in strength of memorized information respectively. Across participants a prioritization account would predict a negative correlation (for the predications based on prioritization and inhibition account see the updated Figure 1). We decided to restrict all analysis to the pre-distractor period now and also only show the analysis collapsed over the entire.5a period. Even though this averaged approach leads to somewhat lower effects in terms of magnitude (e.g. the correlation shown in Figure 4B is lower than in the previous submission in which the relationship was visualised for the greatest statistical effect obtained from the nonparametric cluster analysis), we think that showing the result in this manner leads to a clearer description of the main points.

The outcome of this revise analysis -albeit overall still supporting a prioritization account- has yielded some surprising insights that we missed previously. Importantly it showed that overall desynchronized (low power) states was linked to relatively enhanced strength if memorized information only for β. However, when taking into account the magnitude of power modulation a negative correlation (matching a prioritization account) was only observed for α. Interestingly α power was more strongly modulated within a participant that β. We are not aware of studies showing a similarly differential pattern of auditor cortical α and β oscillations. It will be interesting in future to explore more detail to what extent these differential neural patters map onto different aspects cognitive processed supporting prioritization of memory representations. In order to account for these altered findings we decided to adapt the title accordingly.

The changes are reflected in a novel Figure 4 as well as at multiple points in the text, especially in the description of the results:

“A functional relationship between pre-distractor α / β power and strength of memorized information should be reflected in two manners (with different directions predicted according to a prioritization or inhibition account; see Figure 1B): Firstly, strength of memorized information should differ overall between low and high power bins. And secondly stronger (relative) differences between the bins should be reflected in stronger concomitant differences in strength of memorized information. […] Whereas β desynchronization appears to prioritize memorized information in general, the α processes seem to be dependent on individually varying modulations.”

Furthermore some comments are now added to the Discussion:

“The group-level effect of reduced decoding accuracy prior to the strong distractor could be seen as a support for the first interpretation, however this analysis does not relate the pre-distractor α / β power to estimated strength of memorized information. […] Altogether, our study underlines the value of combining conventional spectral analysis approaches with MVPA in advancing our understanding the functional role of brain oscillations in the auditory system.”

References:

Van Diepen, Rosanne M., John J Foxe, and Ali Mazaheri. „The Functional Role of Α-Band Activity in Attentional Processing: The Current Zeitgeist and Future Outlook. *Current Opinion in Psychology* 29 (Oktober 2019): 229–38. https://doi.org/10.1016/j.copsyc.2019.03.015.